# Posttraumatic Growth and Subjective Well-Being in Men and Women after Divorce: The Mediating and Moderating Roles of Self-Esteem

**DOI:** 10.3390/ijerph20053864

**Published:** 2023-02-22

**Authors:** Kinga Kaleta, Justyna Mróz

**Affiliations:** Department of Psychology, Jan Kochanowski University of Kielce, 25-369 Kielce, Poland

**Keywords:** posttraumatic growth, well-being, self-esteem, divorce, divorcees, gender

## Abstract

Prior research has mainly examined non-adaptive responses to divorce, with less attention being paid to positive changes following the adversity of marital dissolution, especially posttraumatic growth and its consequences. The aim of this paper was to analyse the relationship between posttraumatic growth and subjective well-being, as well as the mediating and moderating role of self-esteem in this relationship among divorced men and women. The sample consisted of 209 divorcees (143 females, 66 males) aged 23–80 (*M* = 41.97, *SD* = 10.72). The Posttraumatic Growth Inventory (PTGI), the Oxford Happiness Questionnaire (OHQ) and the Rosenberg Self-Esteem Scale (SES) were used in the study. Positive associations between overall posttraumatic growth, specific growth dimensions, subjective well-being and self-esteem were found. Self-esteem was confirmed as a mediator in the relationships between changes in perception of self and subjective well-being (SWB), between changes in relating to others and SWB and between appreciation for life and SWB. Self-esteem moderated the association between spiritual changes and subjective well-being; namely, changes in spirituality were positively related to happiness in individuals with lower and average self-esteem but not with high self-esteem. We found no differences between women and men in the obtained results. Self-esteem might be considered a possible psychological (mediating rather than moderating) mechanism in the transmission of PTG onto SWB in divorcees, regardless of their gender.

## 1. Introduction

For decades, marital crisis and divorce have been conceptualised as traumatic events due to exposure to serious psychological harm [1,2,3]. Divorce is a painful relational experience entailing loss of attachment [4], violation of the previous ways of thinking, including disruptions in one’s assumptions about the partner, oneself and the relationship [5], spiritual struggles [6], economic decline, the perceived lack of social support, conflicts with an ex-spouse, problems in co-parenting or loss of custody of children [7]. As a result, divorced individuals might experience posttraumatic stress symptoms of recurrent thoughts, arousal, negative cognition and avoidance behaviour [8]. In the study by Slanbekova et al. [9], almost half of the divorced respondents met the criteria of partial-PTSD or full-PTSD. Numerous studies also showed that the seriously stressful divorce process exacerbated symptoms similar to PTSD, such as considerable stress, anxiety, unhappiness, depression, fear, abandonment, loneliness, guilt, anger, pessimism, decreased activity and efficacy [10,11,12,13,14], as well as vulnerability to diseases, alcohol use and health problems [7,15,16,17]. From this perspective, divorce has been classified as one of the most stressful experiences in adulthood [2]. Moreover, there may be gender differences in the negative consequences of divorce [18,19,20]. For instance, males reported poorer well-being than females in the short term after divorce [19].

Although numerous studies confirmed non-adaptive responses to divorce, scholars have not overlooked positive ones, including resilience, adjustment and beneficial psychological changes, such as greater maturity and growth [21,22,23]. In particular, previous studies showed that divorced individuals reported posttraumatic growth—PTG [24,25] understood as a constructive psychological shift triggered by highly challenging adverse life events [26,27]. The benefits refer to the development of a deeper view of the self, others and the world, improvement of interpersonal relationships, a greater appreciation of life, as well as the development of a more complex meaning, purpose and spirituality [1,28,29]. Although such qualitative changes might occur in a person who experienced the adversity of divorce [21], they are not universal phenomena of a post-divorce adaptation, and a well-adapted person does not necessarily have to experience PTG [13]. In addition, little is known whether and how these positive changes, should they occur, contribute to the well-being of divorced male and female individuals. 

### 1.1. Posttraumatic Growth and Well-Being

The adaptive significance of PTG has been widely accepted, although strong evidence for this hypothesis is still lacking [30]. On the one hand, in previous studies involving various samples, greater benefit finding has been associated with decreased levels of distress, posttraumatic stress disorder symptoms, depression and anxiety [31,32], as well as better mental and physical health [33]. On the other, growth does not necessarily have to result in the reduction of negative symptoms and increased well-being, although they are all dimensions of optimal human functioning [34]. Growth and distress are independent of each other, not the opposite sides of the continuum, and they may coexist [30,35,36,37]. Findings of recent studies, including longitudinal [38] and meta-analytic ones [39,40], confirmed a positive relationship between posttraumatic growth and posttraumatic stress. Thus, researchers can focus their attention either on PTSD-PTG associations, PTSD symptoms, or the PTG itself, examining both their antecedents and consequences. 

Similarly, growth and well-being are distinct concepts, as they have different predictors and outcomes [1,41]. In addition, well-being might be conceptualised in hedonistic tradition as subjective well-being (SWB) or derived from the eudaimonic approach as psychological well-being—PWB [34]. The former is defined as maximising pleasure and avoiding pain; the latter is understood as self-realisation and being a fully functioning person. Conceptually, the constructs of PTG and PWB represent the eudaimonic perspective, as they are both focused on multidimensional human self-actualisation, while SWB refers mainly to an evaluation of life as a whole, life satisfaction, happiness, positive affect and the absence of negative mood [1,34,42,43]. For these reasons, research on the relationships between growth and well-being has given ambiguous findings [43]. Some studies have shown a positive association between growth following adversity and subjective/psychological well-being or quality, satisfaction and meaning in one’s life [44,45,46,47], while others showed no significant relationships [29,36,48,49] if not negative association [50]. McDonough et al. [51] found a positive correlation between PTG and subjective well-being at the point of data collection, but they failed to confirm it three months later. Finally, the studies [43,52] showed that PTG is related to psychological rather than subjective well-being. The above outcomes were, however, received in populations other than divorcees and cannot be simply extended to people who experienced the trauma of marital dissolution and divorce. Therefore, it is important to define precisely what type of well-being, in response to what kind of trauma, is being studied.

Research on PTG among divorcees is sparse and, to a minor extent, explains the relationships between the positive changes and the well-being of these individuals. Particularly, Lamela et al. [1] found no association between PTG and satisfaction with life among divorced individuals. In the study by Krumrei et al. [24], divorcees’ PTG showed a positive correlation with adaptive spiritual coping and a negative correlation with depression. Studley and Chung [53] showed negative correlations between posttraumatic growth following relationship dissolution and psychological comorbidity. Thus, it is still an open question whether PTG is related to well-being among divorced adults, and the first objective of the present study is to examine the relationships between PTG and well-being. In this study, we are especially interested in subjective well-being. It reflects the hedonic path of optimal functioning after a distressing event, especially the levels of an individual’s efficiency in coping with constraints raised by marital dissolution and the ability to balance affective states and achieve satisfaction with life after divorce [34,41,42]. This is what people usually call happiness [54,55], and this approach enables us to find out whether PTG is related to the feeling of happiness after divorce. As PTG was shown to be linked to components of SWB, we hypothesised a positive association between posttraumatic growth and subjective well-being in divorced individuals (H1). However, favourable shifts following divorce may vary across gender [56], and it is important to control for respondents’ gender while examining relationships between PTSD and SWB. 

### 1.2. Possible Role of Self-Esteem

The second goal of the present study is to examine the possible mechanism by which growth and happiness might be associated. According to the affective–cognitive processing theory [57], posttraumatic growth emerges out of a repetitive cyclic process that leads to the resolution of discrepancies between the pre-event assumptions and the trauma-related information. Traumatic events provide information that is incompatible with one’s previous self-view and which initiates processing mechanisms. During such processes, an individual reappraises the event, its impact and meaning in their lives and strives to balance their emotions along with effective acting, which is further cognitively appraised, and the person gets to know that he or she is able to deal with adversity. The feedback cycle, reflective pondering, constructive coping strategies and positive accommodation, should they occur, are expected to be growthful [58,59]. They allow one to rebuild his or her cognitive self-schema by achieving compatibility between pre- and post-event assumptions and perceiving self in a more constructive and positive light [47,57]. These changes in self-perception might reduce negative emotional states and promote positive ones, just as satisfaction with one’s life, commonly summarised as subjective happiness [34,57]. 

Based on the affective–cognitive processing model of PTG, one possible mechanism linking PTG and SWB in divorcees might be increased self-esteem. As PTG refers to the development of a profounder view of the self, the development of a more complex emotional and cognitive regulation system and psychological maturity [1,28,41,60] results in boosted self-esteem [29]. For instance, in a sample of sexually abused women, those who reported benefits from their traumatic event were higher in self-esteem when compared to women who indicated no benefits [61]. Additionally, PTG was a predictor of self-esteem among adolescents who experienced highly stressful life events [62]. Such a relationship might also be assumed in a sample of divorcees who experienced positive psychological changes following marital dissolution. Previous studies exhibited an increase in self-focused growth, sense of coherence, coping skills, self-confidence, optimism and new opportunities and life perspectives as a result of struggling with a distressing event of divorce [14,63,64,65]. Such extension of perceived personal competence drives self-esteem, defined as one’s evaluation of worthiness as an individual [66]. In turn, higher self-esteem maintains positive affect and well-being and facilitates coping with stressors and challenges [67]. Considerable positive associations were reported between self-esteem and subjective well-being measured with Oxford Happiness Inventory [68], as well as between self-esteem and psychological well-being measured with Ryff’s multidimensional PWB scale [69]. As self-esteem was related to both growth and well-being, it might be posited to act as a mediator in the relationship between these variables. Thus, our second hypothesis was formulated in the following way: self-esteem mediates the associations between posttraumatic growth and subjective well-being among divorced individuals (H2). 

On the other hand, a moderating hypothesis is also possible. As stated by Ryan and Deci [34], people high in self-esteem may have attributional styles that are more self-enhancing, which contributes to their happiness. After adversity and following psychological growth, they might benefit from positive changes and experience higher well-being than people with lower self-esteem. In light of this assumption, a moderation model can help explain how the relationship between growth and happiness changes at various levels of self-esteem. Thus, we hypothesised that self-esteem moderates the link between psychological growth and subjective well-being (H3). As self-esteem is gender-specific favouring males [70,71,72,73,74], we used divorced individuals’ gender as a covariate. 

## 2. Materials and Methods

### 2.1. Participants and Data Collection Procedures

Data were obtained from 209 divorced individuals. 

Recruitment to the study was conducted in selected family diagnostic and consultative institutions, including psychological and pedagogical counselling centres, family assistance centres, as well as in schools and family courts. The data was collected in Poland by trained helpers (psychology students recruited for the project) in collaboration with psychologists or pedagogues working in these institutions. The helpers approached about two hundred twenty adults in the above places and recruited respondents who declared themselves as divorcees and gave their consent to take part in the study. The only criterion for inclusion in the study was that the person (woman or man) was divorced. Participants were given a set of questionnaires and asked to complete them at any convenient place and time, usually at their own houses. They were openly informed about the aims of the study and were not paid for completing any measure. Each participant responded to several basic biographical items (e.g., gender, age, time since the divorce) and completed three standardised measures. The study included data from all participants who completed the measures described below.

The participants were 143 divorced females and 66 divorced males aged 23–80 (*M* = 41.97, *SD* = 10.72). Almost half of the participants (49.0%) completed higher education, 26.2% secondary education, 8.6% college education and 16.2% vocational education. They usually lived in cities (50.7%), in towns (31.8%), and less often in the country (17.5%). The majority of the respondents had children (78.4%), one (43.1%), two (43.1%) or more (13.8%), and were employed (74.9%). They mainly evaluated their economic situation as satisfactory, more specifically as average (50.2%) or good and very good (41.6%); however, 7.9% reported poor or very poor financial standing. Time since divorce ranged from 0.5 to 30 years (*M* = 3.26, *SD* = 3.68). The majority of participants were divorce initiators (71.4%) and remained single after divorce (73.7%). 

### 2.2. Measures

#### 2.2.1. *Posttraumatic growth*

Posttraumatic growth was measured using the Polish version of The Posttraumatic Growth Inventory (PTGI) [75,76]. The PTGI contains 21 items (e.g., My priorities in life have changed) which are rated on a 6-point scale, from 0 (I did not experience this change as a result of my crisis) to 5 (I experienced this change to a very great degree as a result of my crisis). The Polish version of PTGI has a four-factor structure. As a result, the PTGI consists of four subscales reflecting transition following adversity in different areas: Changes in self-perception, Changes in relating to others, Appreciation for life, and Spiritual changes. The total PTGI score is obtained by summing scores in the four subscales, and it ranges from 0 to 105. The higher the scores, the greater the PTG. Cronbach’s alpha values in the present study were for the total PTGI 0.96 and for specific subscales 0.94 (Changes in self-perception), 0.92 (Changes in relating to others), 0.88 (Appreciation for life) and 0.80 (Spiritual changes). 

#### 2.2.2. *Subjective Well-Being*

Well-being was assessed with the Polish adaptation [77] of The Oxford Happiness Questionnaire (OHQ) [68]. This version of the tool consists of 26 (shorter version used in this study) or 29 items rated on a 6-point scale ranging from 1 (strongly disagree) to 6 (strongly agree). OHQ enables evaluating satisfaction with one’s own life and self-assurance, as well as the personal resources required for it. The Poprawa’s adaptation [77] has a single-factor structure resulting in one total score, which reflects the level of subjective well-being and potential of happiness. Sample items: I am very happy, I can find beauty in some things, and I make others feel positive. In the current sample, Cronbach’s alpha was 0.89.

#### 2.2.3. *Self-Esteem*

Self-esteem was evaluated using the Polish version [78] of the classic 10-item Rosenberg [79] Self-Esteem Scale (SES). The respondent is requested to rate items on a 4-point scale from 1 (strongly agree) to 4 (strongly disagree). Several items require reverse scoring. Sample items: I feel that I’m a person of worth, and I feel I do not have much to be proud of. A total score ranging from 10 to 40 reflects an individual’s global attitude toward the self. The higher the score, the more positive one’s self-esteem. Cronbach’s alpha for the study sample was 0.89.

### 2.3. Statistical Analysis

First, we computed descriptive statistics of demographic variables for our sample. We then calculated descriptive statistics—means (*M*), standard deviations (*SD*), skewness (*ske*) and kurtosis (*k*)—for all the variables and performed correlational analyses to explore relationships among posttraumatic growth, self-esteem and subjective well-being. Second, to examine whether SES mediates the relationship between PTG and SWB, regression-based mediation analyses were conducted using the Process macro for SPSS (Model 4, version 3.5) [80]. Finally, to test whether SES moderated the relationship between PTG and SWB, we used a set of moderation analyses using the Process macro for SPSS (Model 1, version 3.5) [80]. In order to properly estimate interactions of PTG and SES on SWB, mean centring and standardising were applied. In mediation and moderation analyses, bias-corrected confidence intervals (CI 95%) and the bootstrapping procedure (samples = 5000) were used to calculate direct and indirect effects. Gender, as well as age, education level, time since divorce and initiator status, were included as covariates in each of the mediation and moderation models.

## 3. Results

Table 1 shows the values of the descriptive statistics of the scores obtained on each of the scales for the whole sample. Firstly, it should be noted that the skewness and kurtosis values for all the variables were in the range of 1 to −1, which indicates that the distribution of these variables did not deviate significantly from the normal distribution. To explore the relationships between respondents’ posttraumatic growth, self-esteem and subjective well-being, correlational analyses were performed. Table 1 shows intercorrelations (Pearson’s r) between analysed variables. Gender did not differentiate the results in any of the measures (due to unequal sample sizes, we used the Mann–Whitney U test to examine gender differences). Women and men reported a similar level of posttraumatic growth (*M*_W_ = 62.79, *SD* = 22.21, *M*_M_ = 58.08, *SD* = 22.61, U =11,919.5, *p* = 0.62), subjective well-being (*M*_W_ = 106.04, *SD* = 18.45, *M*_M_ = 100.96, *SD* = 21.97, U = 12,138.5, *p* = 0.65) and self-esteem (*M*_W_ = 28.94, *SD* = 5.12, *M*_M_ = 27.81, *SD* = 5.21, U = 16,845, *p* = 0.52).

As shown, overall PTG and all dimensions of posttraumatic growth were positively linked to subjective well-being. Changes in perception of self, in relating to others and appreciation for life were also positively related to self-esteem, whereas spiritual changes were not linked to self-esteem. There was also a positive link between self-esteem and well-being.

### 3.1. Self-Esteem as a Mediator

To examine whether self-esteem mediated the relationship between posttraumatic growth and subjective well-being, we conducted four mediation analyses. A mediation model has been proposed with four dimensions of PTG as predictors, well-being as an outcome variable, and self-esteem as a mediator (Figure 1). In each of the models, the participants’ gender as well as age, education level, time since divorce and initiator status were included as covariates. The proposed models and regression analyses results, including direct effects, are presented in Figure 1.

Regression analysis showed that changes in self-perception were related to higher self-esteem (B = 0.22; *p* < 0.001), which in turn was related to a higher level of subjective well-being (B = 1.62; *p* < 0.001). A significant indirect effect occurred between self-perception and well-being, with self-esteem being a mediator (IE = 0.36; 95% CI (0.21, 0.53)). The regression analysis revealed that changes in relating to others were associated with higher self-esteem (B = 0.20; *p* < 0.001), which in turn was associated with a higher level of well-being (B = 1.91; *p* < 0.001). There was a significant indirect effect of PTG on happiness through self-esteem (IE = 0.39; 95% CI (0.21, 0.59)). Next, appreciation for life was related to higher self-esteem (B = 0.51; *p* < 0.001), which was related to a higher level of well-being (B = 1.74; *p* < 0.01). A significant indirect effect between the variables through self-esteem was also confirmed (IE = 0.88; 95% CI (0.46, 1.37)). Finally, spiritual changes were not associated with self-esteem (B = 0.01; ns), which in turn was related to higher well-being (B = 2.28; *p* < 0.001). The indirect effect was not significant (IE = 0.03; 95% CI (−0.55, 0.59)). 

All four mediation models were significant and accounted for 40 to 51% of the variance in well-being. They showed a substantial relationship between posttraumatic growth and happiness, and for changes in self-perception, relating to others and appreciation for life, they revealed the mediating role of self-esteem in the effect on subjective well-being. None of the models revealed the gender, age, education level, time since the divorce or initiator status as significant covariates.

### 3.2. Self-Esteem as a Moderator

To test whether self-esteem moderated the relationship between posttraumatic growth and subjective well-being, we used a set of moderation analyses. Moderation models have been proposed with dimensions of PTG as predictors, SWB as an outcome variable, and SES as a moderator. Although all four models were significant and accounted for 42 to 51% of the variance in well-being, three of the four possible interactions turned out to be insignificant, and one was significant. None of the demographic factors, including gender, was a significant covariate in any of the models. 

The analysis revealed no interaction effect of changes in self-perception and self-esteem in relation to well-being (B = 0.02; 95% CI (−0.11; 0.14)). In addition, there were no significant interaction effects of changes in relating to others and self-esteem (B = −0.05; 95% CI (−0.16; 0.07)), nor appreciation for life and self-esteem (B = −0.02; 95% CI (−0.14; 0.10)) on happiness. By contrast, the analysis revealed an interaction effect of spiritual changes and self-esteem in relation to subjective well-being (B = −0.16; 95% CI (−0.28; −0.04), ΔR^2^ = 0.03). Changes in spirituality were positively related to happiness in individuals with a lower (B = 0.30; 95% CI (0.14; 0.46)) and average self-esteem (B = 0.15; 95% CI (0.03; 0.27)), but not with high self-esteem (B = −0.02; 95% CI (−0.20; 0.16)), as shown in Figure 2.

## 4. Discussion

The primary goal of the present study was to analyse the relationship between posttraumatic growth and subjective well-being among divorced individuals. We also aimed to examine the mediating and moderating roles of self-esteem in the hypothesised positive link between PTG and SWB. Our study extends previous research on posttraumatic growth and well-being by demonstrating a positive growth-happiness association in males and females who experienced the adversity of divorce. Moreover, evidence emerged to suggest that self-esteem appears to play a mediating, rather than a moderating, role in the association. Such effects occurred regardless of divorcees’ gender, education level, time since the divorce and initiator status.

Just as we expected, divorcees’ PTG was positively related to SWB. Not only did overall psychological growth correlate with happiness, but also with all its dimensions. The more positive changes in self-perception, in relating to others, appreciation for life and spiritual shifts, the more satisfying functioning in life of those who had experienced a distressing event of divorce. Our results are consistent with the previous studies showing the relationships between benefit finding after adversity and different components of SWB in various samples. They showed, for instance, a positive link between favourable changes and decreased levels of negative emotion, such as anxiety and depressive mood in people who had been severely traumatised [31], and a positive correlation between PTG and mental health life quality and happiness after breast cancer [44]. In the meta-analytic reviews, researchers found that benefit finding following a range of traumas were related to less depression and more positive well-being [36], and PTG after cancer or HIV/AIDS was associated with increased positive mental health (including positive affect and life satisfaction), reduced negative mental health (including depression, anxiety, PTSD) and better subjective physical well-being [81]. PTG was found to be positively associated with life satisfaction in students who had experienced trauma events [45] and in adult non-student participants reporting highly stressful events, including the death of their close loved one, serious medical problems, accidents, physical or sexual assaults and divorce [47]. Finally, PTG was positively related to subjective well-being in women who had experienced intimate partner violence [82]. Our results, just as the results of the above studies, revealed that positive psychological changes after adversity coexist with subjective happiness and life satisfaction. The findings might be interpreted in light of the affective–cognitive processing model of PTG [57]. During the process of growth, cycling changes in one’s cognitions and appraisals lead to the resolution of painful dissonance between pre-trauma and trauma-related information and beliefs. It gradually reduces negative arousal and distress and allows for balancing affective states and feeling satisfied with life, which are components of subjective happiness [34]. 

With regard to our second hypothesis, self-esteem was confirmed as the mediator in the relationships between three dimensions of posttraumatic growth and subjective well-being. Specifically, changes in self-perception, in relation to others, and appreciation for life were related to higher self-esteem, which, in turn, was associated with greater happiness. Self-esteem, since it provides an overall view of the self as less or more worthy, skilled, effective in coping and respected by people [83], can be considered a pathway leading from PTG to SWB. Thus, when people are trying to engage in appropriate cognitive working (deliberate thinking focused on understanding the marital dissolution, finding the meaning of divorce leading to the resolution of core beliefs) and managing their emotions (reducing emotional distress and finding ways to experience positive emotional states), they gain knowledge of themselves as more effective and competent individuals [47,57]. Previous research found coping styles conducive to growth elements, including positive reframing and active coping [36,38,84,85] and positive relationships between constructive coping and self-esteem [86,87,88]. All growth affective–cognitive processes fuel a positive attitude about the self, and when a person feels good about oneself, his or her emotional balance is positive and satisfaction with life is higher. Our results are consistent with the findings of Triplett et al. [47], who revealed PTG having a significant positive effect on life satisfaction, with half being indirect through an increase in meaning in life, which was a result of deliberate rumination and reduced symptoms of distress. Our evidence for the mediating role of self-esteem in the PTG-SWB relationship also allows us to integrate research showing, on the one hand, the positive relationships between PTG and self-esteem [61], and on the other—the positive associations between self-esteem and happiness [68].

We found, however, one exception supporting an alternative moderation model that explains the relationship between spiritual changes and happiness. While self-esteem did not mediate the link between post-divorce spiritual transition and subjective well-being, it was shown to play a moderating role in the association. Changes in spirituality were positively related to happiness in divorcees with lower and average self-esteem but not with high self-esteem. Thus, our third hypothesis about the moderating role of self-esteem in the PTG-SWB link was supported only for one dimension of growth. For individuals with low and medium self-esteem, becoming more spiritually involved after the divorce was positively linked to their well-being. This might suggest that for people who do not rate themselves very highly, spirituality is a source of happiness and life satisfaction. They might use spiritual coping strategies more often than persons high in self-esteem who may rely on themselves rather than on spiritual beliefs and practices. Spiritual coping, in turn, has been shown to be positively related to well-being and life satisfaction after the adversity [89]. 

Taken together, the findings showed that self-esteem operates mainly as a mediator in divorcees’ PTG-SWB link. Self-esteem previously mediated the effects of such positive variables, such as perceived social support, grit and resilience, on life satisfaction [90,91,92]. Moreover, while testing the relationship between loneliness and life satisfaction using self-esteem as a mediator and a moderator, the mediating, but not moderating, role of self-esteem was confirmed [93]. Our findings, just as the findings of prior studies, seem to support the affective–cognitive processing theory [57], demonstrating how psychological growth might enhance the evaluation of oneself that is related to SWB.

Moreover, we found no differences across gender in revealed mechanisms, which requires reference to previous studies. It has been argued that gender differences prior to divorce in variables like self-esteem, attachment or social support are related to gender roles and account for differences in adjustment after the divorce between men and women [94,95]. Some studies also showed gender differences in the after-effects of divorce; for instance, women generated significantly more positive consequences of their divorce than men [18]. However, in the longer term, subjective measures of well-being were similar for females and males, demonstrating that post-divorce adaptation alleviated gender disparities [19]. Our study is consistent with these findings. Other Polish [96,97] studies also failed to find gender as a significant variable in models testing well-being and adjustment after divorce.

### Limitations

Despite offering some promising findings, the study has several limitations. First, the design of the study was cross-sectional, and causation cannot be determined. Although we tested a mediation model in which PTG is related to SWB due to self-esteem, regression-based calculations do not prove an inference that PTG leads to SWB by increased self-esteem. Only a longitudinal study would legitimate such reasoning. Moreover, an alternative model is also possible: higher self-esteem enhances PTG, which in turn increases SWB. According to the terror management theory, self-esteem plays an anxiety-buffering role in the face of threats and adversity [67]. Self-esteem provides the sense of efficacy which is necessary for coping with trauma and maintains positive affect and well-being. Thus, high self-esteem might trigger post-divorce growth that may lead to enhanced subjective well-being. Self-esteem might also be considered a personal resource promoting growth through hardships, as it provides a sense of competence. Abraído-Lanza et al. [98] showed that higher self-esteem positively predicted growth, especially appreciation for life, in over a 3-year long study among individuals facing multiple adversities. Further inquiry should test alternative mediation hypotheses and employ longitudinal studies. Next, there is a significant gender imbalance in the sample, including more females than males. We collected data only from interested individuals who accepted no remuneration. The question is whether some bonuses would encourage more men to participate in the research and help minimise the imbalance. Additionally, most of the participants were highly educated with high incomes, which could have influenced our results. Further research could use a control group of poorer and less educated respondents to test the effect of education and income on PTG or SWB. Then, we only measured SWB, whereas growth was shown to be linked more to psychological than to subjective well-being [43,52]. As SWB and PWB are distinct concepts, they should be measured separately. The inclusion of both types of well-being would make analyses more in-depth and allow to propose more complex models, for instance, with two mediators. Absence of control groups including respondents after different types of trauma (e.g., loss of health, natural disaster, death of loved ones) might be another weakness of the present study, as we do not know whether the findings are typical for individuals specifically after divorce or for individuals experiencing a range of adversity. Finally, the lack of specific data about the nature of the experience of divorce among the participants should also be mentioned as the research deficit. While it is a generally difficult life event, there is great variability in the levels of distress and type of experiences among people. 

## 5. Conclusions

To sum up, the present study extends previous research on PTG in two aspects. First, we found a clear positive association between PTG and SWB in a sample of divorced individuals. Although there is a considerable body of literature showing the perceived benefits after experiencing unhappy life events, such as natural disasters, accidents, physical/sexual assault, combat, bereavement, HIV infection, bone marrow transplantation, chronic illness, cancer, etc. [22], research on growth following the adversity of divorce has been rarely undertaken. Our study is an attempt to fill in this gap. Second, we found that self-esteem is a possible psychological mechanism (mediating rather than moderating) in the transmission of PTG onto SWB. For the relationships between changes in self-perception, in relation to others and appreciation for life, and happiness, self-esteem plays a mediating role, whereas only for spiritual changes—a moderating role.

Turning to practice, the main implication of the above findings is that enhancing PTG is salutary for divorced individuals. Not only does it allow divorcees to find benefits after experiencing adversity, but it also helps recover one’s self-esteem and happiness. The results of the present study might be useful for counselling and psychotherapy in several areas: (a) making divorced clients aware of the possibility of experiencing growth instead of distress or regardless of it, (b) exploring positive life changes following divorce and their relationship with clients’ well-being, (c) strengthening their self-esteem, (d) designing interventions taking into account the importance of posttraumatic growth and self-esteem, and (e) adapting existing therapeutic models for PTG [57,99,100]. They all should be aimed at developing personal strengths, enhancing positive emotions, relationships with important others, and active involvement in one’s life and spiritual values. Such goals might be provided, for instance, by self- and life-affirmation writing, benefit finding, reappraisal, contemplation and meditation, meaning-making and psycho-spiritual interventions, as well as emotion regulation and coping skills training [101].

## Figures and Tables

**Figure 1 ijerph-20-03864-f001:**
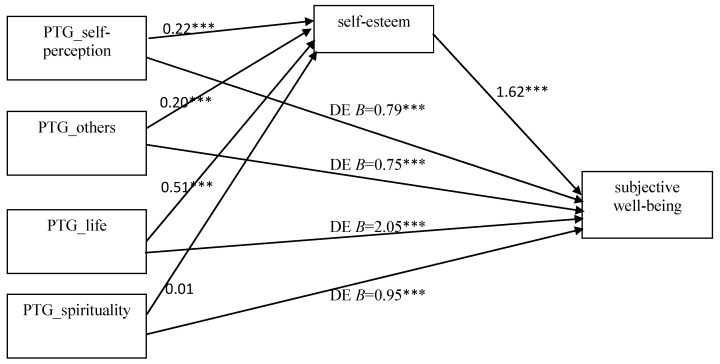
Parallel Mediation Model of Dimensions of PTG on Subjective Well-being Using Self-esteem as a Mediator. Note. Unstandardized coefficients are presented. *** *p* < 0.001.

**Figure 2 ijerph-20-03864-f002:**
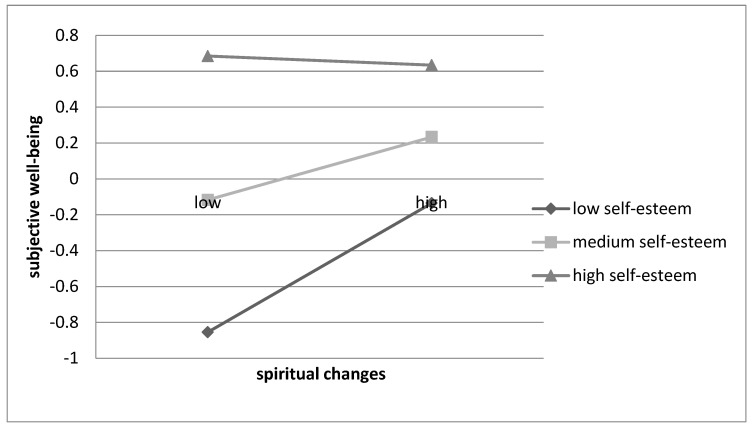
A Graphical Representation of Moderation Effect of Spiritual Changes on Subjective Well-being at Low (−1*SD*), Moderate (*M*) and High (+1*SD*) Levels of Self-esteem.

**Table 1 ijerph-20-03864-t001:** Descriptive Statistics and Pearson’s Correlations between Posttraumatic Growth, Self-esteem, and Subjective Well-being.

Variables	*M*	*SD*	*ske*	*k*	1	2	3	4	5	6	7
1. PTGI Changes in self-perception	28.45	10.61	−0.95	0.55	-						
2. PTG Changes in relating to others	19.62	8.24	−0.43	−0.29	0.71 **	-					
3. PTGI Appreciation for life	9.59	3.92	−0.82	0.05	0.85 **	0.67 **	-				
4. PTGI Spiritual changes	3.74	3.31	0.36	−0.97	0.40 **	0.50 **	0.40 **	-			
5. PTGI total	61.40	22.60	−0.73	0.11	0.93 **	0.89 **	0.87 **	0.58 **	-		
6. Subjective well-being	104.28	19.69	−0.24	−0.42	0.60 **	0.47 **	0.57 **	0.16 *	0.58 **	-	
7. Self-esteem	28.56	5.40	−0.36	0.26	0.45 **	0.33 **	0.39 **	0.01	0.40 **	0.61 **	-

* *p* < 0.05, ** *p* < 0.01.

## Data Availability

The dataset presented in this study is available on reasonable request from the corresponding author.

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
