# Peer review of "Posttraumatic Growth and Subjective Well-Being in Men and Women after Divorce: The Mediating and Moderating Roles of Self-Esteem"

_ijerph, 2023, doi:10.3390/ijerph20053864_

Round 1

Reviewer 1 Report

Thank you for the opportunity to review the original manuscript titled ‘Post-traumatic growth and subjective well-being in men and women after divorce: The mediating and moderating roles of self-esteem’. This paper aimed to analyse the relationships between PTG and wellbeing, and self-esteem in divorcees. Please see my comments below.

Comments:

1.      Sex and gender are used interchangeably, I think you are referring to gender so please keep it consistent.

2.      Introduction is thorough and well written.

3.      Were the helpers research assistants or students?

4.      Does active in the labor market mean they were employed or looking for employment?

5.      Results are clear and reasonable

6.      Most of your participants were highly educated with high income, do you think this impacted results? And did you ask about cultural backgrounds?

7.      Do you also think that because most of your participants initiated the divorce that this protected their wellbeing? And surely if someone did not initiate the divorce, this would impact their self-esteem?

Reviewer 2 Report

In the attached file

Reviewer 3 Report

I think this article is very interesting and important for psychological science and practice. I would recommend it for publication after a minor revision:

1.     In the abstract, the authors go straight from the purpose of the study to the results, skipping the methods. I think it would be better if the authors added to the abstract the instruments and a description of the study sample, indicating the number of respondents.

2.     The authors used divorced individuals’ sex as a covariate, referring to the fact that self-esteem is a gender-specific phenomenon. Meanwhile, self-esteem also changes with age (e.g., Ogihara, Y., & Kusumi, T. (2020). The developmental trajectory of self-esteem across the life span in Japan: Age differences in scores on the Rosenberg Self-Esteem Scale from adolescence to old age. Frontiers in Public Health, 8, 132. https://doi.org/10.3389/fpubh.2020.00132). I would like to understand why the authors did not use age as another covariate.

3.     It is unclear to me why the authors wrote that all of the study variables did not deviate significantly from the normal distribution, but used Mann-Whitney U test to compare posttraumatic growth scores in men and women. In addition, the authors did not justify the choice of comparison criterion in the statistical analysis.
